# Psychometric Testing of the CEECCA Questionnaire to Assess Ability to Communicate among Individuals with Aphasia

**DOI:** 10.3390/ijerph20053935

**Published:** 2023-02-22

**Authors:** Willian-Jesús Martín-Dorta, Alfonso-Miguel García-Hernández, Jonathan Delgado-Hernández, Estela Sainz-Fregel, Raquel-Candelaria Miranda-Martín, Alejandra Suárez-Pérez, Alejandra Jiménez-Álvarez, Elena Martín-Felipe, Pedro-Ruymán Brito-Brito

**Affiliations:** 1Primary Care Management of Tenerife, The Canary Islands Health Service, 38400 Puerto de la Cruz, Spain; 2Department of Nursing, University of La Laguna, 38200 San Cristóbal de La Laguna, Spain; 3CREN Specialist Rehabilitation Centre, 38296 San Cristóbal de La Laguna, Spain; 4Rehabilitation Department, University Hospital of the Canary Islands, The Canary Islands Health Service, 38320 San Cristóbal de La Laguna, Spain; 5Neurology Department, Nuestra Señora de la Candelaria University Hospital, The Canary Islands Health Service, 38010 Santa Cruz de Tenerife, Spain; 6Training and Research in Care, Primary Care Management Board of Tenerife, The Canary Islands Health Service, Department of Nursing, University of La Laguna, 38200 San Cristóbal de La Laguna, Spain

**Keywords:** aphasia, surveys and questionnaires, standardised nursing terminology, nursing assessment, validation studies

## Abstract

(1) Background: The CEECCA questionnaire assesses the ability to communicate among individuals with aphasia. It was designed using the NANDA-I and NOC standardised nursing languages (SNLs), reaching high content validity index and representativeness index values. The questionnaire was pilot-tested, demonstrating its feasibility for use by nurses in any healthcare setting. This study aims to identify the psychometric properties of this instrument. (2) Methods: 47 individuals with aphasia were recruited from primary and specialist care facilities. The instrument was tested for construct validity and criterion validity, reliability, internal consistency, and responsiveness. The NANDA-I and NOC SNLs and the Boston test were used for criterion validity testing. (3) Results: five language dimensions explained 78.6% of the total variance. Convergent criterion validity tests showed concordances of up to 94% (Cohen’s *κ*: 0.9; *p* < 0.001) using the Boston test, concordances of up to 81% using DCs of NANDA-I diagnoses (Cohen’s *κ*: 0.6; *p* < 0.001), and concordances of up to 96% (Cohen’s *κ*: 0.9; *p* < 0.001) using NOC indicators. The internal consistency (Cronbach’s alpha) was 0.98. Reliability tests revealed test–retest concordances of 76–100% (*p* < 0.001). (4) Conclusions: the CEECCA is an easy-to-use, valid, and reliable instrument to assess the ability to communicate among individuals with aphasia.

## 1. Introduction

Aphasia is a multimodal cognitive disorder caused by acquired brain damage that impacts spoken language, listening comprehension, reading, writing, and other cognitive processes dependent on the activity of the left cerebral hemisphere, which are essential for the proper functioning of language [1,2]. Cerebrovascular accidents (CVAs) are the most common cause of aphasia [3]. Multicentre studies show an incidence of aphasia ranging from 18% to 38% [4,5,6,7].

The use of language is one of the most distinctive attributes of the human species, enabling us to become social beings and partake in a particular culture [8]. Through language, we interact with those around us and ourselves, and we learn and organise our thoughts. Language represents a basic feature of our personality. Given the various functions that language has for human beings, its impairment or loss has a significant impact on the quality of life of individuals with aphasia, making it difficult for them to carry out certain activities of daily living and negatively affecting their physical, mental, emotional, familial, and social wellbeing [9,10,11]. The consequences of aphasia can be exacerbated by inappropriate communication strategies in the healthcare setting [12]. It has been shown that when nurses interact with individuals with aphasia, patients are given little opportunity to start or maintain a conversation, and conversations revolve around the professional’s goals rather than the patient’s needs and demands [13]. This asymmetrical communication leads to a loss of autonomy and makes it difficult for the patient with aphasia to participate in their own care, resulting in feelings of frustration, helplessness, fear, anger, and/or resignation [14].

The complexity of aphasia requires a multidisciplinary and interdisciplinary approach, in which nurses have a fundamental role to play as care providers. Assessment, the first step in the nursing process, is key to ensuring an accurate diagnosis and a care plan tailored to the needs of the individual with aphasia [15]. We believe that this assessment should also facilitate the search for strategies to achieve the most symmetrical and effective nurse–patient communicative interaction possible. The growing production of nursing studies in the past two decades reflects the interest of researchers in caring for patients with language and communication disorders [16,17]. However, surprisingly few studies have been devoted to the assessment of the communicative abilities of aphasia patients. Most aphasia assessment tests have emerged from disciplines such as neurology, psychology, linguistics, and speech therapy. Exploratory modalities include screening tests designed to determine the presence or absence of aphasia; assessment batteries, commonly extensive tests constructed from multiple subtests usually requiring specific knowledge for the interpretation of results; and tests to assess one specific component of language in particular, such as naming or language comprehension [18,19,20]. The characteristics of these assessment tools make them unsuitable for daily use by nurses [19].

The CEECCA, Cuestionario para la Evaluación Enfermera de las Capacidades Comunicativas en la Afasia (Nursing Assessment of Ability to Communicate among Patients with Aphasia questionnaire), is based on the NANDA-I nursing diagnoses classification (NANDA-I), the Nursing Outcomes Classification (NOC), and the pictograms of the ARASAAC, Centro Aragonés para la Comunicación Aumentativa y Alternativa (Aragonese Centre for Alternative and Augmentative Communication). The CEECCA is intended to assess the communicative abilities among patients with aphasia based on areas of interest for care. It consists of 43 items assessing 14 specific areas of language, corresponding to five global dimensions likely to be affected in aphasia: “Verbal expression”, “Written expression”, “Expression through pictograms”, “Auditory comprehension”, and “Reading comprehension”. Its design includes the defining characteristics (DCs) of the two communication-related NANDA-I nursing diagnoses: “Impaired verbal communication” (00051) and “Readiness for enhanced communication” (00157), as well as the NOC outcome indicators “Communication” (0902), “Communication: expressive” (0903), “Communication: receptive” (0904), and “Information processing” (0907). In an initial study [21], the design and construction process of the CEECCA and the results of the content validity tests were published. In these tests, the CEECCA obtained high content validity and representativeness scores. This study demonstrated the utility of the NANDA-I and NOC classifications for the construction of instruments to improve the accuracy of nursing diagnosis and the measurement of outcome indicators in specific care settings. After this preliminary study, it was necessary to determine the remaining psychometric properties regarding the validity and reliability of the CEECCA instrument in a sample of individuals with aphasia.

The research hypothesis of this study was that the CEECCA is a valid and reliable instrument for nurses to use to assess the ability to communicate among individuals with aphasia, including dimensions of interest for care.

The objectives of this study were to carry out the necessary validity and reliability tests to obtain the psychometric properties of the CEECCA (criterion validity, construct validity, internal consistency, and reliability) and to describe the profile of the patients with aphasia in the validation sample based on their sociodemographic and clinical variables.

## 2. Materials and Methods

A study of design and validation of a health questionnaire was proposed. The protocol used was based on the proposals put forward by Carvajal et al. [22] and Ramada-Rodilla et al. [23] for validating health measurement instruments already used in other studies on the design and validation of assessment instruments based on the NANDA-I and NOC classifications [24,25]. Once the design, construction, and content validity phases of the CEECCA were completed [21], the instrument was tested on a sample of individuals with aphasia to calculate the remaining psychometric properties. Data collection process and statistical tests used to determine the validity and reliability of the questionnaire are described below.

### 2.1. Data Collection

This stage consisted of four phases.
(A)Selecting the members of the research team. Three nurses were selected using a convenience sampling method: two from the Primary Care Management Board of the Tenerife Healthcare Area and one from the Management Board of the Nuestra Señora de La Candelaria University Hospital. The instruction phase for the professionals comprised five joint explanatory meetings of approximately 60 min each. Their content focused on explaining the objectives and methodology of the study, the instructions for administering the CEECCA and the proxy instruments, as well as other methodological and ethical considerations. Each collaborator was given a field notebook with all the documents required for the administration of the tests and data collection.(B)Selecting the sample. Participants were selected using a convenience sampling method in various settings: The University Hospital of the Canary Islands (HUC), primary healthcare facilities in the Tenerife Healthcare Area, as well as private rehabilitation centres and associations. Inclusion criteria were persons aged 18 and over, with an active diagnosis of aphasia registered in the clinical record as a consequence of acquired brain damage, with aphasic symptoms detected by The Boston test for diagnosing aphasia (Spanish adaptation, second edition) [26], with Spanish as their mother tongue, and who agreed to participate in the study. The exclusion criteria were patients with a low level of consciousness (in a vegetative state and/or minimally conscious), a personal history of neurological or neurodegenerative disease prior to the brain injury that caused the aphasia, a psychiatric–psychological history of communication disorder prior to the brain damage, a cognitive level preventing them from taking the test, pre-morbid reading and writing disability, severe visual or hearing impairment that hinders the correct administration of the instrument, behavioural problems that impede collaboration with the researchers, or a history of alcoholism and/or other drug abuse.(C)Administering the proxy instruments. The three proxy instruments used in the validation phase of the CEECCA questionnaire had previously been used in the design and construction phase of the questionnaire, enabling consistency to be maintained between the two processes: The Boston test for diagnosing aphasia (Spanish adaptation, second edition) [26]; the selected indicators of the four communication-related Nursing Outcomes Classification (NOC) outcomes [27]; and the selected defining characteristics (DCs) of the 2012–2014 NANDA-I nursing diagnosis “Impaired verbal communication” [28]. The standardised nursing classifications used were the latest revisions available at the time of administration.

Firstly, a speech therapist administered the activities selected from the Boston test for the diagnosis of aphasia:Conversational speech.Descriptive speech.Visual confrontation naming.Writing mechanics.Written confrontation naming.Auditory discrimination of words.Auditory comprehension of commands.Reading comprehension. Matching pictures and words.Reading comprehension. Reading sentences and paragraphs.

Additionally, this test corroborated the diagnosis of aphasia registered in the clinical record.

During this first visit, clinical and sociodemographic data were collected and informed consent was obtained.

At an interval of one to three days, a nurse assessed the selected communication-related NOC outcome indicators and determined the presence or absence of the DCs of the NANDA-I diagnosis “Impaired verbal communication”.
(D)A nurse administered the CEECCA to each subject in the sample. The CEECCA was administered at the primary healthcare facilities in the Tenerife Healthcare Area, at the HUC rehabilitation units, in the rehabilitation departments of the collaborating centres and associations, and in the participants’ own homes. The CEECCA was administered once more by another nurse under the same conditions at an interval of one to seven days. Alternatively, one of the nurses who previously administered the questionnaire repeated the process four weeks later.

### 2.2. Data Analysis

The results obtained from the administration of the CEECCA questionnaire and the proxy instruments, and the data on clinical and sociodemographic variables, were entered as they became available into an SPSS v.25.0 database for further refinement and processing. The data processing plan involved four phases.

#### 2.2.1. Sample Size

The necessary sample size was calculated by taking as a reference the sample sizes used to validate other instruments included in a 2017 systematic review aimed at identifying and evaluating the psychometric properties of screening-type tests for diagnosing post-stroke aphasia [29]. The review included nine studies [30,31,32,33,34,35,36,37,38]. The samples of aphasia patients, with whom these tests were validated, had an average of 42 participants. Taking this data as a reference and considering the difficulty in recruiting participants with this clinical and psychosocial profile, a sample of around 50 participants was deemed necessary to estimate the correlation coefficients for analysing the convergent criterion validity and reliability coefficients (Cohen’s *κ*) of the questionnaire through non-random concordance estimates of at least 0.30 while maintaining 95% confidence levels. The sample was described by expressing nominal variables as absolute and relative frequencies and by expressing quantitative variables as the median (minimum–maximum).

#### 2.2.2. Validity Tests

Construct validity

Validity tests were carried out using a principal component analysis, following the Kaiser–Meyer–Olkin (KMO) sampling adequacy measure and Bartlett’s test of sphericity, confirming the dimensions that make up the questionnaire, using a varimax rotation to check that the component items of the questionnaire load towards the areas that theoretically make up its dimensions. The analysis was performed using the scores obtained from each of the subjects in the sample in the first administration of the CEECCA.

Convergent criterion validity

Concordances between the first administration of the CEECCA questionnaire and the three proxy instruments were estimated using Cohen’s *κ* corrected for random chance effects. Each area of the CEECCA was compared with the selected areas of the Boston test, as shown in Table 1.

To obtain the concordance between the two instruments, the results of each area of the CEECCA were used as dichotomous variables (i.e., functional/dysfunctional) in accordance with the qualitative rules designed for this purpose. It was agreed to select the 60th and 70th percentiles from the percentile table summarising the results of the Boston test. Two concordance tests were performed with this proxy instrument, with results equal to or above the selected percentiles being considered functional, while results below them were considered dysfunctional.

Regarding the NANDA-I classification [28], Cohen’s *κ* concordance degrees were estimated between the results of each area of the CEECCA (in terms of functionality/dysfunctionality) and the presence of DCs of the NANDA-I diagnosis “Impaired verbal communication” relating to that particular area (Table 2).

The degree of concordance between the results of each area in the CEECCA and the results of the evaluation through the four communication-related NOC outcome indicators [27] was also calculated (Table 3). For this proxy instrument, two concordance tests were performed: an initial test, in which scores from 3 to 5 on the Likert scale inclusive, and assessing each NOC outcome indicator, were set as the criterion for functionality; and a second test, with scores of 4 and 5 considered functional. This conversion allowed the NOC indicator scores to be reformulated into dichotomous variables. Cohen’s *κ* statistic was again used.

Additionally, non-parametric correlations were calculated between the CEECCA total scores and the total scores of the selected subtests in the Boston test, the total number of DCs of the NANDA-I diagnostic label, and the total score obtained in the evaluation of the NOC outcome indicators for the sample. The Spearman–Brown rho statistic was used for this calculation.

#### 2.2.3. Inter-Observer Reliability, Intra-Observer Reliability, and Internal Consistency

The degree of concordance between nurses (with an interval between one and seven days) and of individual nurses (with a four-week interval) was calculated for the results of each area in the CEECCA. These estimates were made using Cohen’s *κ* concordance statistic corrected for random chance effects. As a supplementary reliability analysis, internal consistency tests were carried out by calculating Cronbach’s alpha and the correlation between each CEECCA item with the other component items of the instrument. This was calculated using the Spearman–Brown rho statistic.

#### 2.2.4. Responsiveness

The responsiveness of the questionnaire was tested on a sample subject who underwent a two-week, 20-hour intensive speech therapy rehabilitation programme. The two-hour sessions were held over a period of five days. The intervention consisted of conversation therapy supplemented with activities of increasing difficulty that focused on the subject’s affected processes. The intervention was conducted by a speech therapist with expertise in this type of intervention. The CEECCA was administered two days before the intervention and again the day after the end of the intervention. In addition, the selected subtests of the Boston test were administered before and after the intervention to check for changes using a benchmark instrument.

## 3. Results

### 3.1. Sample Description

The sample consisted 47 subjects diagnosed with aphasia, with 16 females (34%) and 31 males (66%) recruited from 20 May 2019 to 18 February 2020 (9 months and 5 days). Their median age was 65 years (41–94 years). All participants were recruited on the island of Tenerife, in the Canary Islands, Spain. Thirty-four percent of the subjects were recruited in primary care consultations, 44.7% in specialised care consultations, and 21.3% in other associations or rehabilitation centres. Regarding their level of education, 25.5% could read and write, 34% had completed primary education, 12.8% had completed secondary education, 17% had a technical or vocational training degree, and 10.6% had a university-level education. Chronic health problems were present in 93.6% of the sample. The most prevalent comorbidities were high blood pressure (72.3%), dyslipidaemia (57.4%), depression (36.2%), urinary incontinence (27.7%), atrial fibrillation (23.4%), obesity (21.3%), constipation (21.3%), epilepsy (19.1%), type 2 diabetes mellitus (17%), anxiety (12.8%), faecal incontinence (10.6%), dysphagia (10.6%), and insomnia (10.6%). Aetiological factors leading to aphasia included ischaemic stroke (59.6%), haemorrhagic stroke (17%), neurodegenerative disease (10.6%), traumatic brain injury (TBI) (6.4%), central nervous system infection (2.1%), and brain tumour (2.1%). The types of aphasia in their clinical records included anomic aphasia (25.5%), mixed transcortical aphasia (19.1%), global aphasia (17%), primary progressive aphasia (12.8%), motor aphasia (4.3%), transcortical motor aphasia (4.3%), anomic motor aphasia (4.3%), semantic variant primary progressive aphasia (4.3%), and transcortical sensory aphasia (2.1%). One participant in the sample (2.1%) could not be assigned any of the established aphasic syndromes. Chronic aphasia with a course of more than 12 months was present in 76.5% of the sample. The majority of the sample was right-handed (97.9%), with only one participant being left-handed.

### 3.2. Administration of the CEECCA

The mean duration of the first administration of the questionnaire was 16 min (9–32), the second administration was 15 min (5–37), and the third administration was 15 min (8–37).

After the first administration of the CEECCA, the language area with the highest percentage of dysfunctionality was “Naming actions in writing” (72.3%), followed by “Verbal expression: descriptive speech”, “Naming objects verbally”, and “Auditory comprehension of sentences” (each with 57.4%). The areas with the lowest percentages of dysfunctionality were “Expressing actions through pictograms” (14.9%) and “Auditory comprehension of words” (17.0%). The remaining areas displayed percentages of dysfunctionality between 44.7% and 29.8%.

### 3.3. Construct Validity

Barlett’s test of sphericity provided a result of 903 (*p* < 0.001), and the Kaiser–Meyer–Olkin (KMO) statistic was 0.30. The five theoretical dimensions explained 78.6% of the total variance. Table 4 shows the rotated component matrix describing the grouping of the items in the five dimensions.

Given the strong correspondence between the theoretical locations of the component items of the CEECCA and the statistical locations resulting from the factor analysis, the decision was made not to make any changes to the initial structure of the instrument.

### 3.4. Convergent Criterion Validity

Table 5 shows Cohen’s *κ* correlation coefficients comparing each area of the CEECCA with the selected areas of the Boston test [26], and taking the 70th and 60th percentiles as references.

Table 6 shows the results of the convergent criterion validity tests comparing each CEECCA area with the selected DCs of the NANDA-I diagnosis “Impaired verbal communication” [28].

Table 7 shows the results of the convergent criterion validity tests, comparing each area of the CEECCA with the selected indicators of the four NOC outcomes related to communication [27].

The non-parametric correlations between the CEECCA score and the scores of each of the proxy instruments for the whole sample are shown below (Table 8).

### 3.5. Reliability through Internal Consistency

The internal consistency value (Cronbach’s alpha) was 0.98. The intensity of the strength of the inter-item correlation is represented using different colours in Figure 1 [40].

### 3.6. Test–Retest Reliability

#### 3.6.1. Inter-Nurse Reliability in Administering the CEECCA

Table 9 shows the inter-nurse reliability results in terms of functionality and dysfunctionality for each area of the questionnaire in the first two administrations, with a time interval of one to seven days.

#### 3.6.2. Intra-Nurse Reliability When Administering the CEECCA

The reliability results in terms of functionality and dysfunctionality for each area of the CEECCA when the same nurse administered the questionnaire at baseline and at one month are shown below. Table 10 shows the results for nurse (a) and Table 11 shows the results for nurse (b).

### 3.7. Responsiveness

The CEECCA areas that exhibited the greatest changes after the intervention (i.e., from a dysfunctional to a functional outcome) were “Descriptive speech”, “Naming objects verbally”, “Writing name and surname(s)”, “Naming objects in writing”, “Naming actions in writing”, and “Auditory comprehension of sentences”. The areas that did not change in terms of functionality but obtained better scores after the intervention were “Conversational speech” and “Naming actions verbally”. The areas that remained unchanged after the intervention were “Expressing actions through pictograms”, “Expressing emotions through pictograms”, “Auditory comprehension of words”, “Auditory comprehension of verbal commands”, “Reading comprehension of words”, and “Reading comprehension of sentences”. No item in the CEECCA areas displayed poorer scores after the intervention. The Boston test subtests obtained better scores, with only the scores on the subtest “Conversational speech” remaining unchanged.

The resulting CEECCA questionnaire is available in Appendix A.

## 4. Discussion

The psychometric tests carried out on the assessment instrument derived from this study have yielded satisfactory results, providing a valid, reliable tool for nurses to assess the main dimensions of language in individuals with aphasia in a simple way and adapted to their daily work. The CEECCA is an instrument whose design [21] and validation processes incorporate aspects of NANDA-I nursing diagnoses and NOC outcome criteria. This allows consistency to be maintained throughout the nursing process as applied in clinical practice [41].

A potential limitation of this study is the sample size used to validate this instrument. In our opinion, the time constraints and operational limitations of the study, together with the difficulty in recruiting participants with this clinical and neuropsychological profile, have prevented a larger sample size from being recruited. Other nursing assessment instruments based on the NANDA-I and NOC classifications have been validated using larger sample sizes [24,25,42]. However, sample sizes are notably smaller in several validation studies of screening-type instruments for the diagnosis of aphasia [30,31,32,33,36,37,38]. These studies do not discuss the reasons for using such a limited sample of subjects with aphasia; however, the frequency of this phenomenon suggests that other studies have also encountered difficulties in recruiting subjects fitting this profile. Similarly, limited samples of patients with aphasia were reported in other studies not devoted to the design and validation of language assessment instruments. For instance, in 2020 a systematic review [43] on the use of transcranial direct current stimulation (tDCS) and a speech therapy intervention in patients with aphasia illustrates this point. This review included 35 studies, with a mean sample size of 14 participants and only one study with more than 40 participants. Some studies mentioned the challenge of obtaining informed consent from individuals with language and communication disorders, resulting in a systematic exclusion of people with aphasia from the samples of many studies [44,45]. A Cochrane review assessing the effectiveness of different strategies in improving the care provided to post-stroke patients and their families [46] revealed that, of the 14 reviewed studies, only one included patients with aphasia, and ten studies considered the presence of aphasia as an exclusion criterion. The authors believe that, in future CEECCA reviews, a larger sample size for validation should be a priority, along with a longer research period.

The calculation of the construct validity of the questionnaire began with carrying out the sampling adequacy tests that warranted the performance of a factor analysis. Bartlett’s test of sphericity, with statistical significance being *p* < 0.05, indicated that the variables that made up the test were correlated and, therefore, a factor analysis could be performed. However, the Kaiser–Meyer–Olkin (KMO) statistic provided a result of 0.30, suggesting that the data fitted a factor model poorly [47,48], which was mainly due to the limited sample size. As a result, the calculation of the total variance explained with five dimensions gave a result of 78.6%, a high value that points to the possibility of reducing the dimensions of the questionnaire, as three dimensions explained 72.0% of the total variance. However, it was decided not to reduce the number of dimensions and to explore the statistical locations of the items. The rationale for this decision was in the interest of maintaining a questionnaire structure and design that would allow the diagnostic labels of dysfunctionality to be established for the dimensions derived from the selection process based on the NANDA-I and NOC classifications and screening instruments for the diagnosis of aphasia. The rotated component matrix distributed the items into their different factors, maintaining a similar and coherent structure to the one proposed in theory. Despite this, most of the items assessing the “Auditory comprehension” dimension, especially the “Auditory comprehension of words” area, shared a factor with the items assessing the “Expression through pictograms” dimension. In this regard, the CEECCA uses a multiple-choice auditory word recognition test to assess these areas. The two tasks necessarily involve the same processes; therefore, it was expected that a patient with dysfunctional auditory comprehension of words assessed using the CEECCA will perform relatively poorly in the area of “Expression through pictograms”. 

Convergent criterion validity tests using the Boston Diagnostic Aphasia Examination (second edition) as a proxy instrument showed concordance percentages representing moderate to strong correlations for most of the areas compared using both the 60th and 70th percentile as references [39,49]. The areas in the CEECCA questionnaire with the strongest correlation were “Conversational speech”, “Descriptive speech”, “Naming objects verbally”, “Naming actions in writing”, and “Auditory comprehension of verbal commands”, with total concordance percentages of up to 93.7%. The area with the lowest kappa value was “Auditory comprehension of words”, with a total percentage of less than 60% and a weak correlation for both the 60th percentile and the 70th percentile. These results may be explained by the different methods used by the two tools for assessing this area. While the CEECCA assesses this area using an auditory discrimination test with five very familiar words with high levels of agreement in terms of naming, the Boston test assesses this area using 36 words from six semantic categories, with different levels of phonemic complexity, lexical frequency, and imaginability. The Boston test will, therefore, be able to identify problems in the discrimination of less familiar words and will be more sensitive in detecting impairment in the recognition of words belonging to a particular semantic class. However, the comprehensive assessment of the patient’s performance in all language areas of the Boston test makes it an instrument that requires a long administration time (between one and a half to two hours) [35] and specific knowledge on the part of the assessor in order to make a proper evaluation of the patient [50]. In turn, the CEECCA seeks opportunities for communicative interaction in each language area through a simple assessment process that does not require a long administration time. To this end, it was necessary to limit the number of items and prioritise the interest in detecting functionality/dysfunctionality in patients with more severe communication disorders, even knowing the loss of sensitivity that the tool would experience in identifying dysfunctionality in milder or more selective communicative disorders. Another aspect to consider is that the Boston test was administered by speech therapists with experience in the care of individuals with aphasia, while the CEECCA was administered by nurses without specific knowledge in speech rehabilitation. Even so, the percentages of total concordance between the two tests ranged between 93.7% (*p* < 0.001) and 55.3% (*p* < 0.001) for the 70th percentile, and between 91.5% (*p* < 0.001) and 57.4% (*p* < 0.001) for the 60th percentile. These data were supported by the degree of correlation between the Boston test subtest total scores and total performance on the CEECCA questionnaire as measured using the Spearman–Brown correlation coefficient, with a coefficient of 0.96 (*p* < 0.001) indicating a strong positive association [51].

Convergent criterion validity tests that used the presence or absence of the DCs of the NANDA-I diagnosis “Impaired verbal communication” as correlation variables indicated *κ* values suggesting weak to moderate concordance strengths [39,49]. On this point, it is important to mention that the NANDA-I classification is not a diagnostic tool, and, therefore, it may be questionable to perform a criterion validity test using it. However, we believe that it is interesting to consider the possibility that the diagnostic labels proposed by the CEECCA serve as sublevels of specificity of the diagnostic labelling proposed by the NANDA-I. In this test, it was observed that the DCs that were more specifically related to the area of language assessed had higher concordance strengths. For example, the language area “Naming objects verbally” related to seven DCs of the NANDA-I diagnosis. In this case, the percentages of total concordance with the DCs “Difficulty forming words” and “Slurred speech” were higher than with the DC “Difficulty expressing thoughts verbally”, which refers to a manifestation not necessarily related to a verbal naming problem. When calculating the strength of the correlation between the CEECCA total scores and the total scores for the DCs of the NANDA-I diagnosis present in the sample, the results indicated a strong negative correlation of −0.85 (*p* < 0.001). This negative correlation was due to the assignment of a value between zero (the poorest possible response) and four (or three) (the best possible CEECCA response), so that a lower CEECCA score for the whole sample was correlated with a higher number of DCs present in the sample.

The *κ* values obtained in the convergent criterion validity tests using the selected indicators of the four communication-related NOC outcomes indicated moderate to strong correlations [39,49]. When a score of 1 or 2 on the Likert scale of the NOC taxonomy was considered dysfunctional, 82% of concordances observed were moderate to very strong; when taking a score of 1, 2, or 3 as dysfunctional, the percentage of moderate to very strong concordances dropped to 60%. The Spearman–Brown correlation coefficient showed a strong positive correlation (0.91) between the CEECCA scores and the NOC indicator assessment scores for the whole sample. A study on the psychometric properties of an instrument (CoNOCidietDiabetes) [25], whose design and validation used the NOC classification, also obtained better levels of correlation between the total scores of the two instruments (r_s_ = 0.72; *p*-value = 0.001) than between the individual correlations for each NOC indicator, where 41% of correlations were rated as weak and only 9.1% were rated as moderate. Other studies on the design and validation of instruments based on the NOC indicators obtained values similar to those obtained in this study when comparing the results for the whole sample using conceptually similar instruments as a reference. For instance, a 2015 study [52] evaluating the psychometric properties of an instrument reported that the Spanish version of a pain level scale based on the NOC outcome “Pain level” showed a strong correlation (r_s_ = −0.81; *p*-value < 0.001) with the numerical pain rating scale.

In general, convergent criterion validity tests for the CEECCA questionnaire appeared to show adequate levels, which were even higher than those obtained in other design and validation studies based on the NANDA-I and NOC classifications with larger sample sizes [25,42,52].

The internal consistency of the questionnaire was high, with a Cronbach’s alpha value of 0.98 and a predominance of moderate and strong inter-item correlations. Although such a high value may suggest item redundancy, Cronbach’s alpha value does not increase when an item is removed from the questionnaire. This, together with our interest in maintaining a structure that would allow diagnostic labels of dysfunctionality to be established for each area of the CEECCA, meant that a reduction in the number of items was not considered.

Inter-nurse reliability when administering the CEECCA (i.e., when two different nurses administer the instrument in an interval of one to seven days) showed concordance percentages above 90%, with *κ* values above 0.75 (*p* < 0.001).

Intra-nurse reliability, both when it was the same nurse in the first and third administration (nurse a) and in the second and third administration (nurse b), showed concordance percentages above 80% in twelve of the fourteen areas of the questionnaire. The areas with the lowest levels of concordance were the areas corresponding to comprehension-related dimensions, especially the “Auditory comprehension of sentences and verbal commands” and “Expression through pictograms” dimensions. Goodglass and Kaplan [26] pointed out that test–retest reliability should be interpreted with caution due to the high fluctuations in the performance of patients with aphasia; however, they also note that, when aphasia becomes chronic, variability in language performance is markedly reduced. On the other hand, a time interval of four weeks does not seem to be long enough to explain this change as an effect of the progress of the disorder itself, or to explain a significant change because of rehabilitation if the patient was receiving it. In the reviewed literature, there is no clear consensus on the most appropriate time interval for conducting an intra-rater reliability test on subjects with aphasia. An interval of 20 to 40 days has been used to calculate intra-rater reliability in other similar studies with subject samples without aphasia [53,54,55]. For the Community Integration Questionnaire Adjusted for People with Aphasia, only inter-rater reliability testing was performed, not intra-rater reliability testing [56]. None of these test–retest reliability calculations were performed for other instruments such as the Frenchay Aphasia Screening Test [30] or the Ullevaal Aphasia Screening (UAS) test [32].

The fact that a high percentage of the sample (66.2%) was receiving speech therapy rehabilitation at the time of assessment could be considered as a change variable for the results at one month; however, in more than 76% of the sample, aphasia had been present for more than 15 months. This reduces the likelihood of relevant changes caused by rehabilitation in a 4-week interval. In addition, none of the rehabilitation interventions received during this stage underwent changes in their characteristics or intensity.

To test the responsiveness of the CEECCA, a specific intervention was introduced that modifies the intensity and characteristics of the speech therapy rehabilitation. The intervention consisted of an intensive therapy based on conversation therapy, supplemented with activities of increasing difficulty focusing on the affected processes, lasting 20 h and spread over ten sessions. Although the available evidence is not yet sufficient to determine, categorically, at what time intervals and intensity levels positive results occur, some authors suggest that a minimum of two hours per day, for a period of two to three weeks, can be considered intensive treatment [57]. The intensity of rehabilitation treatment is considered a relevant variable for its success [20]. A number of systematic reviews assessing the effects of speech and communication therapy in patients with post-stroke aphasia [58,59] highlight the positive relationship between high-intensity rehabilitation treatments and improved outcomes in functional communication and writing. In recent years, a growing number of studies point in the same direction, concluding that high-intensity rehabilitative interventions improve learning and brain plasticity and strengthen synaptic contacts between neurons [60,61,62], even among patients with chronic aphasia [63,64]. The subject assessed in this test was diagnosed with motor aphasia with a course of more than four years. According to the pre-intervention assessment, the areas relating to the verbal and written expression dimensions exhibited the greatest degree of dysfunctionality. After the intervention, these same areas showed the greatest improvements when compared to their baseline levels, according to both the CEECCA and the Boston test.

## 5. Conclusions

The preliminary results obtained suggest that the CEECCA is a valid, reliable instrument for the nursing assessment of the ability of individuals with aphasia to communicate, including dimensions of interest for their care. Using the GRAQoL Index, which assesses the psychometric properties of health measurement instruments through the fulfilment of set criteria, the CEECCA questionnaire obtained a final score of 75%, with an A grade of recommendation and above average results when compared to other health instruments [65]. The CEECCA can be administered at any stage of aphasia and in any healthcare setting. This instrument favours nurse–patient communication by indicating which dimensions and areas of language are functional in order to maintain a communicative exchange.

## Figures and Tables

**Figure 1 ijerph-20-03935-f001:**
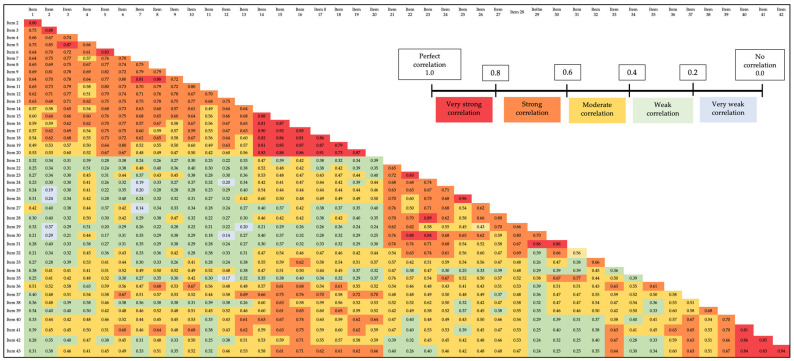
Inter-item correlation matrix rated according to the strength of correlation (Spearman’s rho) [40].

**Table 1 ijerph-20-03935-t001:** Correspondence between areas of the CEECCA and areas of the Boston test to calculate the convergent criterion validity.

Areas in the CEECCA Questionnaire	Areas in the Boston Test
- CEECCA. Conversational speech.	- Boston. Conversational speech.
- CEECCA. Descriptive speech.	- Boston. Descriptive speech.
- CEECCA. Naming objects verbally. - CEECCA. Naming actions verbally.	- Boston. Visual confrontation naming.
- CEECCA. Writing name and surname(s).	- Boston. Writing mechanics.
- CEECCA. Naming objects in writing. - CEECCA. Naming actions in writing.	- Boston. Written confrontation naming.
- CEECCA. Auditory comprehension of words.	- Boston. Auditory discrimination of words.
- CEECCA. Auditory comprehension of sentences. - CEECCA. Auditory comprehension of verbal commands.	- Boston. Auditory comprehension of commands.
- CEECCA. Reading comprehension of words.	- Boston. Reading comprehension. Matching pictures and words.
- CEECCA. Reading comprehension of sentences.	- Reading comprehension. Reading sentences and paragraphs.

**Table 2 ijerph-20-03935-t002:** Correspondence between the areas in the CEECCA and the DCs of the NANDA-I diagnosis “Impaired verbal communication” to calculate the convergent criterion validity.

Areas in the CEECCA Questionnaire	Defining Characteristics. Impaired Verbal Communication (2015–2017)
- CEECCA. Conversational speech. - CEECCA. Descriptive speech.	- Difficulty expressing thoughts verbally (e.g., aphasia, dysphasia, apraxia, dyslexia). - Difficulty forming sentences. - Difficulty forming words (e.g., aphonia, dyslalia, dysarthria). - Difficulty speaking. - Difficulty maintaining communication. - Difficulty verbalising. - Slurred speech. - Inappropriate verbalisation.
- CEECCA. Naming objects verbally. - CEECCA. Naming actions verbally.	- Difficulty expressing thoughts verbally (e.g., aphasia, dysphasia, apraxia, dyslexia). - Difficulty forming words (e.g., aphonia, dyslalia, dysarthria). - Difficulty speaking. - Difficulty maintaining communication. - Difficulty verbalising. - Slurred speech. - Inappropriate verbalisation.
- CEECCA. Writing name and surname(s). - CEECCA. Naming objects in writing. - CEECCA. Naming actions in writing.	- Difficulty expressing thoughts verbally (e.g., aphasia, dysphasia, apraxia, dyslexia). - Difficulty maintaining communication.
- CEECCA. Expressing actions through pictograms. - CEECCA. Expressing emotions through pictograms.	- Difficulty comprehending communication. - Difficulty in use of body expressions.
- CEECCA. Auditory comprehension of words. - CEECCA. Auditory comprehension of sentences. - CEECCA. Auditory comprehension of verbal commands.	- Difficulty comprehending communication. - Difficulty maintaining communication.
- CEECCA. Reading comprehension of words. - CEECCA. Reading comprehension of sentences.	- Difficulty comprehending communication.

**Table 3 ijerph-20-03935-t003:** Correspondence between the areas in the CEECCA and the selected indicators of the four communication-related NOC outcomes to calculate the convergent criterion validity.

Areas in the CEECCA Questionnaire	NOC Outcome. Indicator
- CEECCA. Conversational speech.	- Communication. Indicator 2 - Communication. Indicator 8 - Communication. Indicator 9 - Communication: expressive. Indicator 2 - Communication: expressive. Indicator 3 - Information processing. Indicator 2 - Information processing. Indicator 9 - Information processing. Indicator 10
- CEECCA. Descriptive speech.	- Communication. Indicator 2 - Communication. Indicator 8 - Communication. Indicator 9 - Communication: expressive. Indicator 2 - Communication: expressive. Indicator 3 - Information processing. Indicator 2 - Information processing. Indicator 3 - Information processing. Indicator 4 - Information processing. Indicator 9 - Information processing. Indicator 10
- CEECCA. Naming objects verbally. - CEECCA. Naming actions verbally.	- Communication. Indicator 2 - Communication. Indicator 8 - Communication. Indicator 9 - Communication: expressive. Indicator 2 - Communication: expressive. Indicator 3 - Information processing. Indicator 1 - Information processing. Indicator 2
- CEECCA. Writing name and surname(s).	- Communication. Indicator 1 - Communication: expressive. Indicator 1
- CEECCA. Naming objects in writing. - CEECCA. Naming actions in writing.	- Communication. Indicator 1 - Communication: expressive. Indicator 1 - Information processing. Indicator 1
- CEECCA. Expressing actions through pictograms. - CEECCA. Expressing emotions through pictograms.	- Communication. Indicator 3 - Communication. Indicator 6 - Communication. Indicator 7 - Communication: expressive. Indicator 4 - Communication: receptive. Indicator 3 - Communication: receptive. Indicator 5 - Communication: receptive. Indicator 6 - Information processing. Indicator 1
- CEECCA. Auditory comprehension of words.	- Communication. Indicator 6 - Communication. Indicator 7 - Communication. Indicator 8 - Communication: receptive. Indicator 2 - Communication: receptive. Indicator 6 - Information processing. Indicator 1
- CEECCA. Auditory comprehension of sentences. - CEECCA. Auditory comprehension of verbal commands.	- Communication. Indicator 6 - Communication. Indicator 7 - Communication. Indicator 8 - Communication: receptive. Indicator 2 - Communication: receptive. Indicator 6 - Information processing. Indicator 5 - Information processing. Indicator 6 - Information processing. Indicator 7
- CEECCA. Reading comprehension of words.	- Communication. Indicator 1 - Communication. Indicator 6 - Communication. Indicator 7 - Communication: receptive. Indicator 1 - Information processing. Indicator 1
- CEECCA. Reading comprehension of sentences.	- Communication. Indicator 1 - Communication. Indicator 6 - Communication. Indicator 7 - Communication: receptive. Indicator 1 - Information processing. Indicator 5 - Information processing. Indicator 6 - Information processing. Indicator 7

**Table 4 ijerph-20-03935-t004:** CEECCA rotated component matrix.

Items in the CEECCA	Components				
	1	2	3	4	5
Conversational speech. Item 3	0.87				
Conversational speech. Item 5	0.86				
Conversational speech. Item 2	0.85				
Naming actions verbally. Item 11	0.85				
Naming objects verbally. Item 9	0.84				
Naming objects verbally. Item 8	0.82				
Naming objects verbally. Item 10	0.82				
Naming objects verbally. Item 7	0.82				
Naming actions verbally. Item 12	0.79				
Conversational speech. Item 1	0.77				
Naming actions verbally. Item 13	0.74				
Descriptive speech. Item 6	0.73				
Conversational speech. Item 4	0.66				
Auditory comprehension of verbal commands. Item 36	0.50				
Auditory comprehension of words. Item 30		0.90			
Auditory comprehension of words. Item 31		0.87			
Expressing actions through pictograms. Item 21		0.83			
Expressing actions through pictograms. Item 23		0.81			
Expressing actions through pictograms. Item 22		0.79			
Auditory comprehension of words. Item 28		0.79			
Auditory comprehension of words. Item 29		0.77			
Expressing emotions through pictograms. Item 24		0.76			
Auditory comprehension of verbal commands. Item 35		0.74			
Auditory comprehension of words. Item 27		0.74			
Expressing emotions through pictograms. Item 25		0.71			
Expressing emotions through pictograms. Item 26		0.70			
Auditory comprehension of sentences. Item 32		0.62			
Naming actions in writing. Item 20			0.75		
Naming objects in writing. Item 17			0.74		
Naming actions in writing. Item 19			0.71		
Writing name and surname(s). Item 14			0.71		
Naming objects in writing. Item 15			0.67		
Naming actions in writing. Item 18			0.59		
Naming objects in writing. Item 16			0.58		
Auditory comprehension of verbal commands. Item 37			0.45		
Reading comprehension of sentences. Item 42				0.88	
Reading comprehension of sentences. Item 43				0.85	
Reading comprehension of sentences. Item 41				0.79	
Reading comprehension of words. Item 40				0.76	
Auditory comprehension of sentences. Item 33				0.52	
Reading comprehension of words. Item 39					0.55
Reading comprehension of words. Item 38					0.55
Auditory comprehension of sentences. Item 34					0.50

Extraction method: principal component analysis. Rotation method: Varimax with Kaiser normalisation.

**Table 5 ijerph-20-03935-t005:** Cohen’s κ correlation coefficients between the CEECCA and the Boston test taking the 70th and 60th percentiles as references.

CEECCA (Areas)	Boston Test (Areas)	70th Percentile					60th Percentile				
		Functionality Concordance (%)	Dysfunctionality Concordance (%)	Total Concordance (%)	Cohen’s *κ* Value	*p*-Value	Functionality Concordance (%)	Dysfunctionality Concordance (%)	Total Concordance (%)	Cohen’s *κ* Value	*p*-Value
Conversational speech	Conversational speech	48.9	29.8	78.7	0.57 *	<0.001	61.7	29.8	91.5	0.81 ***	<0.001
Descriptive speech	Descriptive speech	42.6	51.1	93.7	0.87 ***	<0.001	42.6	40.4	83	0.67 **	<0.001
Naming objects verbally	Visual confrontation naming	25.5	44.7	70.2	0.43 *	<0.001	34.0	44.7	78.7	0.49 *	<0.001
Naming actions verbally	Visual confrontation naming	25.5	38.3	63.8	0.35	0.002	34.0	38.3	72.3	0.49 *	<0.001
Writing name and surname(s)	Writing mechanics: name	17.0	44.7	61.7	0.26	0.017	44.7	44.7	89.4	0.79 **	<0.001
Naming objects in writing	Written confrontation naming	31.9	55.3	87.2	0.73 **	<0.001	32.6	50.0	82.6	0.64 **	<0.001
Naming actions in writing	Written confrontation naming	25.5	59.6	85.1	0.67 **	<0.001	26.1	54.3	80.4	0.59 *	<0.001
Auditory comprehension of words	Auditory discrimination of words	38.3	17.0	55.3	0.23	0.014	40.4	17.0	57.4	0.24	0.011
Auditory comprehension of sentences	Auditory comprehension of verbal commands	29.8	40.4	70.2	0.42 *	0.001	36.2	36.2	72.4	0.45 *	0.001
Auditory comprehension of verbal commands	Auditory comprehension of commands	34.0	44.7	78.7	0.59 *	<0.001	40.4	40.4	80.8	0.62 **	<0.001
Reading comprehension of words	Reading comprehension: matching pictures and words	25.5	38.3	63.8	0.35	0.002	34.0	38.3	72.3	0.49 *	<0.001
Reading comprehension of sentences	Reading comprehension: reading sentences and paragraphs.	31.9	55.3	87.2	0.73 *	<0.001	31.9	55.3	87.2	0.73 **	<0.001

Level of agreement according to Landis and Koch [39]: <0.20 slight; 0.21–0.40 fair; 0.41–0.60 moderate *; 0.61–0.80 substantial **; 0.81–1.00 almost perfect ***.

**Table 6 ijerph-20-03935-t006:** Cohen’s *κ* correlation coefficients between the CEECCA areas and the DCs of the NANDA-I diagnosis “Impaired verbal communication”.

CEECCA (Area)	NANDA-I Diagnosis Impaired Verbal Communication (DCs)	Functionality Concordance (%)	Dysfunctionality Concordance (%)	Total Concordance (%)	Cohen’s *κ* Value	Sig. *p*
Conversational speech	DC2. Difficulty expressing thoughts verbally (e.g., aphasia, dysphasia, apraxia, dyslexia).	8.5	31.9	40.4	0.08	0.152
	DC3. Difficulty forming sentences.	21.3	31.9	53.2	0.23	0.015
	DC4. Difficulty forming words (e.g., aphonia, dyslalia, dysarthria).	34.0	31.9	65.9	0.39	0.001
	DC5. Difficulty speaking.	23.4	29.8	53.2	0.21	0.042
	DC6. Difficulty maintaining communication	23.4	31.9	55.3	0.25	0.009
	DC9. Difficulty verbalising.	23.4	29.8	53.2	0.21	0.042
	DC13. Slurred speech.	36.2	29.8	66.0	0.37	0.002
	DC14. Inappropriate verbalisation.	31.9	31.9	63.8	0.36	0.001
Descriptive speech	DC2. Difficulty expressing thoughts verbally (e.g., aphasia, dysphasia, apraxia, dyslexia).	8.5	57.4	65.9	0.22	0.015
	DC3. Difficulty forming sentences.	19.1	55.3	74.4	0.44 *	0.001
	DC4. Difficulty forming words (e.g., aphonia, dyslalia, dysarthria).	27.7	51.1	78.8	0.55 *	<0.001
	DC5. Difficulty speaking.	19.1	51.1	70.2	0.36	0.008
	DC6. Difficulty maintaining communication.	23.4	57.4	80.8	0.58 *	<0.001
	DC9. Difficulty verbalising.	14.9	46.8	61.7	0.17	0.200
	DC13. Slurred speech.	27.7	46.8	74.5	0.47 *	0.001
	DC14. Inappropriate verbalisation.	27.7	53.2	80.9	0.60 *	<0.001
Naming objects verbally	DC2. Difficulty expressing thoughts verbally (e.g., aphasia, dysphasia, apraxia, dyslexia).	8.5	44.7	53.2	0.14	0.060
	DC4. Difficulty forming words (e.g., aphonia, dyslalia, dysarthria).	31.9	42.6	76.6	0.51 *	<0.001
	DC5. Difficulty speaking.	23.4	42.6	66.0	0.35	0.003
	DC6. Difficulty maintaining communication.	23.4	44.7	68.1	0.40	0.001
	DC9. Difficulty verbalising.	23.4	42.6	66.0	0.35	0.003
	DC13. Slurred speech.	34.0	40.4	74.4	0.50 *	<0.001
	DC14. Inappropriate verbalisation.	29.8	42.6	72.4	0.47 *	<0.001
Naming actions verbally	DC2. Difficulty expressing thoughts verbally (e.g., aphasia, dysphasia, apraxia, dyslexia).	8.5	38.3	46.8	0.11	0.099
	DC4. Difficulty forming words (e.g., aphonia, dyslalia, dysarthria).	34.0	38.3	72.3	0.49 *	<0.001
	DC5. Difficulty speaking.	25.5	38.3	63.8	0.35	0.002
	DC6. Difficulty maintaining communication.	23.4	38.4	61.8	0.32	0.003
	DC9. Difficulty verbalising.	23.4	36.2	59.6	0.28	0.013
	DC13. Slurred speech.	36.2	36.2	72.4	0.48 *	<0.001
	DC14. Inappropriate verbalisation.	31.9	38.3	70.2	0.45 *	<0.001
Writing name and surname(s)	DC2. Difficulty expressing thoughts verbally (e.g., aphasia, dysphasia, apraxia, dyslexia).	6.4	44.7	51.1	0.07	0.361
	DC6. Difficulty maintaining communication.	19.1	42.6	61.7	0.26	0.300
Naming objects in writing	DC2. Difficulty expressing thoughts verbally (e.g., aphasia, dysphasia, apraxia, dyslexia).	6.4	59.6	66.0	0.16	0.114
	DC6.Difficulty maintaining communication.	17.0	55.3	72.3	0.37	0.007
Naming actions in writing	DC2. Difficulty expressing thoughts verbally (e.g., aphasia, dysphasia, apraxia, dyslexia).	6.4	70.2	76.6	0.26	0.027
	DC6. Difficulty maintaining communication.	12.8	61.7	74.5	0.33	0.023
Expressing actions through pictograms	DC1. Difficulty comprehending communication.	44.7	14.9	59.6	0.25	0.010
	DC7. Difficulty in use of body expressions.	48.9	12.8	61.7	0.22	0.035
Expressing emotions through pictograms	DC1. Difficulty comprehending communication.	38.3	23.4	61.7	0.27	0.037
	DC7. Difficulty in use of body expressions.	42.6	21.3	63.9	0.27	0.045
Auditory comprehension of words	DC1. Difficulty comprehending communication.	44.7	17.0	61.7	0.28	0.005
	DC6. Difficulty maintaining communication.	23.4	17.0	40.4	0.12	0.086
Auditory comprehension of sentences	DC1. Difficulty comprehending communication.	38.3	38.3	76.6	0.54 *	<0.001
	DC6. Difficulty maintaining communication.	21.3	42.6	63.9	0.32	0.007
Auditory comprehension of verbal commands	DC1. Difficulty comprehending communication.	38.3	38.3	76.6	0.54 *	<0.001
	DC6. Difficulty maintaining communication.	21.3	42.6	63.9	0.31	0.007
Reading comprehension of words	DC1. Difficulty comprehending communication.	42.6	36.2	78.8	0.59 *	<0.001
Reading comprehension of sentences	DC1. Difficulty comprehending communication.	31.9	46.8	78.7	0.57 *	<0.001

Level of agreement according to Landis and Koch [39]: <0.20 slight; 0.21–0.40 fair; 0.41–0.60 moderate *; 0.61–0.80 substantial; 0.81–1.00 almost perfect.

**Table 7 ijerph-20-03935-t007:** Cohen’s *κ* correlation coefficients between the CEECCA and NOC outcome indicators.

CEECCA (Areas)	NOC Outcomes. Indicators	When a Score of 1 or 2 on the Likert Scale of the NOC Taxonomy Was Considered Dysfunctional					When a Score of 1, 2, or 3 on the Likert Scale of the NOC Taxonomy Was Considered Dysfunctional				
		Functionality Concordance (%)	Dysfunctionality Concordance (%)	Total Concordance (%)	Cohen’s *κ* Value	Sig. *p*	Functionality Concordance (%)	Dysfunctionality Concordance (%)	Total Concordance (%)	Cohen’s *κ* Value	Sig. *p*
Conversational speech	Communication. Indicator 2	46.8	29.8	76.6	0.54 *	<0.001	31.9	31.9	63.8	0.36	0.001
	Communication. Indicator 8	51.1	31.9	83.0	0.66 **	<0.001	29.8	31.9	61.7	0.33	0.002
	Communication. Indicator 9	51.1	29.8	80.9	0.61 **	<0.001	36.2	31.9	68.1	0.42 *	<0.001
	Communication: expressive. Indicator 2	46.8	29.8	76.6	0.54 *	<0.001	31.9	29.8	61.7	0.31	0.007
	Communication: expressive. Indicator 3	46.8	31.9	78.7	0.58 *	<0.001	29.8	31.9	61.7	0.33	0.002
	Information processing. Indicator 2	51.1	31.9	83.0	0.66 **	<0.001	25.5	31.9	57.4	0.28	0.006
	Information processing. Indicator 9	46.8	29.8	76.6	0.54 *	<0.001	31.9	31.9	63.8	0.36	0.001
	Information processing. Indicator 10	46.8	31.9	78.7	0.58 *	<0.001	31.9	31.9	63.8	0.36	0.001
Descriptive speech	Communication. Indicator 2	40.4	48.9	89.3	0.79 **	<0.001	31.9	57.4	89.3	0.78 **	<0.001
	Communication. Indicator 8	40.4	46.8	87.2	0.75 **	<0.001	27.7	55.3	83	0.64 **	<0.001
	Communication. Indicator 9	40.4	44.7	85.1	0.71 **	<0.001	31.9	53.2	85.1	0.69 **	<0.001
	Communication: expressive. Indicator 2	31.9	55.3	87.2	0.73 **	<0.001	40.4	51.1	91.5	0.83 ***	<0.001
	Communication: expressive. Indicator 3	29.8	57.4	87.2	0.73 **	<0.001	42.6	48.9	91.5	0.83 ***	<0.001
	Information processing. Indicator 2	42.6	48.9	91.5	0.83 ***	<0.001	25.5	57.4	82.9	0.63 **	<0.001
	Information processing. Indicator 3	42.6	10.6	53.2	0.16	0.042	40.4	48.9	89.3	0.79 **	<0.001
	Information processing. Indicator 4	42.6	8.5	51.1	0.13	0.072	40.4	48.9	89.3	0.79 **	<0.001
	Information processing. Indicator 9	40..4	48.9	89.3	0.79 **	<0.001	29.8	55.3	85.1	0.69 **	<0.001
	Information processing. Indicator 10	40.4	51.1	91.5	0.83 ***	<0.001	29.8	55.3	85.1	0.69 **	<0.001
Naming objects verbally	Communication. Indicator 2	46.8	42.6	89.4	0.79 **	<0.001	31.9	44.7	76.6	0.55 *	<0.001
	Communication. Indicator 8	48.9	42.6	92.5	0.83 ***	<0.001	29.8	44.7	74.5	0.51 *	<0.001
	Communication. Indicator 9	48.9	40.4	89.3	0.79 **	<0.001	36.2	44.7	80.9	0.63 **	<0.001
	Communication: expressive. Indicator 2	48.9	44.7	93.6	0.87 ***	<0.001	34.0	44.7	78.7	0.59 *	<0.001
	Communication: expressive. Indicator 3	46.8	44.7	91.5	0.83 ***	<0.001	29.8	44.7	74.5	0.51 *	<0.001
	Information processing. Indicator 1	53.2	19.1	72.3	0.41 *	0.001	36.2	42.6	78.8	0.59 *	<0.001
	Information processing. Indicator 2	51.1	44.7	95.8	0.92 ***	<0.001	25.5	44.7	70.2	0.43 *	<0.001
Naming actions verbally	Communication. Indicator 2	48.9	38.3	87.2	0.75 **	<0.001	31.9	38.3	70.2	0.45 *	<0.001
	Communication. Indicator 8	46.8	34.0	80.8	0.62 **	<0.001	29.8	38.3	68.1	0.42 *	<0.001
	Communication. Indicator 9	51.1	36.2	87.3	0.74 **	<0.001	36.2	38.3	74.5	0.52 *	<0.001
	Communication: expressive. Indicator 2	48.9	38.3	87.2	0.75 **	<0.001	34.0	38.3	72.3	0.49 *	<0.001
	Communication: expressive. Indicator 3	46.8	38.3	85.1	0.71 **	<0.001	29.8	38.3	68.1	0.74 **	<0.001
	Information processing. Indicator 1	57.4	17.0	74.4	0.41 *	0.002	36.2	36.2	72.4	0.48 *	<0.001
	Information processing. Indicator 2	48.9	12.8	61.7	0.70 **	<0.001	25.5	38.3	63.8	0.35	<0.001
Writing name and surname(s)	Communication. Indicator 1	36.2	46.8	83.0	0.67 **	<0.001	21.3	46.8	68.1	0.38	<0.001
	Communication: expressive. Indicator 1	31.9	46.8	78.7	0.58 *	<0.001	21.3	46.8	68.1	0.38	<0.001
Naming objects in writing	Communication. Indicator 1	31.9	57.4	89.3	0.77 **	<0.001	21.3	61.7	83.0	0.61 **	<0.001
	Communication: expressive. Indicator 1	27.7	57.4	85.1	0.68 **	<0.001	21.3	61.7	83.0	0.61 **	<0.001
	Information processing. Indicator 1	34.0	17.0	51.0	0.14	0.180	29.8	53.2	83.0	0.64 **	<0.001
Naming actions in writing	Communication. Indicator 1	23.4	59.6	83.0	0.61 **	<0.001	19.1	70.2	89.3	0.71 **	<0.001
	Communication: expressive. Indicator 1	23.4	63.8	87.2	0.70 **	<0.001	19.1	70.2	89.3	0.71 **	<0.001
	Communication: receptive. Indicator 3	27.7	21.3	49.0	0.19	0.028	25.5	59.6	85.1	0.67 **	<0.001
Expressing actions through pictograms	Communication. Indicator 3	57.4	10.6	68.0	0.24	0.051	38.3	14.9	53.2	0.20	0.024
	Communication. Indicator 6	74.5	10.6	85.1	0.50 *	<0.001	51.1	14.9	66	0.31	0.003
	Communication. Indicator 7	66.0	10.6	76.6	0.35	0.009	38.3	14.9	53.2	0.20	0.024
	Communication: expressive. Indicator 4	55.3	10.6	65.9	0.21	0.070	34.0	14.9	48.9	0.17	0.039
	Communication: receptive. Indicator 3	72.3	10.6	82.9	0.46 *	0.001	55.3	12.8	68.1	0.29	0.012
	Communication: receptive. Indicator 5	80.9	8.5	89.4	0.56 *	<0.001	63.8	10.6	74.4	0.32	0.015
	Communication: receptive. Indicator 6	68.1	12.8	80.9	0.47 *	<0.001	46.8	14.9	61.7	0.27	0.007
	Information processing: Indicator 1	78.7	14.9	93.6	0.79 **	<0.001	38.3	14.9	53.2	0.20	0.024
Expressing emotions through pictograms	Communication. Indicator 3	51.1	19.1	70.2	0.34	0.017	36.2	27.7	63.9	0.34	0.004
	Communication. Indicator 6	63.8	14.9	78.7	0.45 *	0.002	42.6	21.3	63.9	0.27	0.045
	Communication. Indicator 7	57.4	17.0	74.4	0.39	0.008	34.0	25.5	59.5	0.26	0.027
	Communication: expressive. Indicator 4	48.9	19.1	68.0	0.31	0.030	29.8	25.5	55.3	0.21	0.063
	Communication: receptive. Indicator 3	61.7	14.9	76.6	0.40	0.005	48.9	21.3	70.2	0.37	0.009
	Communication: receptive. Indicator 5	68.1	10.6	78.7	0.39	0.002	55.3	17.0	72.3	0.35	0.016
	Communication: receptive. Indicator 6	57.4	17.0	74.4	0.39	0.008	42.6	25.5	68.1	0.38	0.004
	Information processing. Indicator 1	63.8	14.9	78.7	0.45 *	0.002	34.0	25.5	59.5	0.26	0.027
Auditory comprehension of words	Communication. Indicator 6	76.6	14.9	91.5	0.73 **	<0.001	51.1	17.0	68.1	0.35	0.002
	Communication. Indicator 7	68.1	14.9	83.0	0.54 *	<0.001	38.3	17.0	55.3	0.23	0.014
	Communication. Indicator 8	51.1	17.0	68.1	0.35	0.002	29.8	17.0	46.8	0.16	0.043
	Communication: receptive. Indicator 2	72.3	17.0	89.3	0.70 **	<0.001	42.6	17.0	59.6	0.26	0.008
	Communication: receptive. Indicator 6	70.2	17.0	87.2	0.65 **	<0.001	46.8	17.0	63.8	0.31	0.004
	Communication: receptive. Indicator 1	78.7	17.0	95.7	0.86 ***	<0.001	38.3	17.0	55.3	0.23	0.014
Auditory comprehension of sentences	Communication. Indicator 6	51.1	17.0	68.1	0.32	0.011	42.6	36.2	78.8	0.57 *	<0.001
	Communication. Indicator 7	51.1	25.5	76.6	0.51 *	<0.001	36.2	42.6	78.8	0.59 *	<0.001
	Communication. Indicator 8	40.4	34.0	74.4	0.49 *	0.001	27.7	42.6	70.3	0.43 *	0.001
	Communication: receptive. Indicator 2	48.9	21.3	70.2	0.38	0.006	38.3	40.4	78.7	0.58 *	<0.001
	Communication: receptive. Indicator 6	46.8	21.3	68.1	0.33	0.016	38.3	36.2	74.5	0.49 *	0.001
	Information processing. Indicator 5	51.1	31.9	83.0	0.65 **	<0.001	38.3	42.6	80.9	0.63 **	<0.001
	Information processing. Indicator 6	44.7	34.0	78.7	0.57 *	<0.001	36.2	42.6	78.8	0.59 *	<0.001
	Information processing. Indicator 7	42.6	36.2	78.8	0.57 *	<0.001	27.7	42.6	70.3	0.43 *	0.001
	Communication. Indicator 6	55.3	21.3	76.6	0.50 *	<0.001	42.6	48.9	91.5	0.57 *	<0.001
	Communication. Indicator 7	53.2	27.7	80.9	0.60 *	<0.001	31.9	38.3	70.2	0.42 *	0.002
	Communication. Indicator 8	46.8	40.4	87.2	0.74 **	<0.001	27.7	42.6	70.3	0.43 *	0.001
	Communication: receptive. Indicator 2	53.2	25.5	78.7	0.55 *	<0.001	38.3	40.4	78.7	0.58 *	<0.001
	Communication: receptive. Indicator 6	53.2	27.7	80.9	0.60 *	<0.001	38.3	36.2	74.5	0.49 *	0.001
	Information processing. Indicator 5	53.2	34.0	87.2	0.74 **	<0.001	38.3	42.6	80.9	0.63 **	<0.001
	Information processing. Indicator 6	48.9	38.3	87.2	0.74 **	<0.001	36.2	42.6	78.8	0.59 *	<0.001
	Information processing. Indicator 7	46.8	40.4	87.2	0.74 **	<0.001	27.7	42.6	70.3	0.43 *	<0.001
Reading comprehension of words	Communication. Indicator 1	36.2	38.3	74.5	0.52 *	<0.001	21.3	38.3	59.6	0.29	0.005
	Communication. Indicator 6	59.6	19.1	78.7	0.51 *	<0.001	46.8	34.0	80.8	0.62 **	<0.001
	Communication. Indicator 7	57.4	25.5	82.9	0.62 **	<0.001	34.0	34.0	68.0	0.40	0.003
	Communication: receptive. Indicator 1	46.8	36.2	83.0	0.66 **	<0.001	25.5	38.3	63.8	0.35	0.002
	Information processing. Indicator 1	59.6	19.1	78.7	0.51 *	<0.001	38.3	38.3	76.6	0.56 *	<0.001
Reading comprehension of sentences	Communication. Indicator 1	25.5	48.9	74.4	0.46 *	0.002	17.0	55.3	72.3	0.38	0.004
	Communication. Indicator 6	40.4	21.3	61.7	0.31	0.003	31.9	40.4	72.3	0.45 *	0.002
	Communication. Indicator 7	40.4	29.8	70.2	0.45 *	<0.001	27.7	51.1	78.8	0.55 *	<0.001
	Communication: receptive. Indicator 1	34.0	44.7	78.7	0.57 *	<0.001	23.4	57.4	80.8	0.58 *	<0.001
	Information processing. Indicator 5	36.2	31.9	68.1	0.39	0.003	29.8	48.9	78.7	0.56 *	<0.001
	Information processing. Indicator 6	36.2	40.4	76.6	0.54 *	<0.001	29.8	51.1	80.9	0.60 *	<0.001
	Information processing. Indicator 7	34.0	42.6	76.6	0.53 *	<0.001	21.3	51.1	72.4	0.41 *	0.005

Level of agreement according to Landis and Koch [39]: <0.20 slight; 0.21–0.40 fair; 0.41–0.60 moderate *; 0.61–0.80 substantial **; 0.81–1.00 almost perfect ***.

**Table 8 ijerph-20-03935-t008:** Non-parametric correlations between the CEECCA and proxy instruments for the whole sample (Spearman’s rho).

Spearman’s Rho		BOSTON_SUM	NANDACDSUM	NOCINDSUM
CEESUMTOT	Correlation coefficient	0.914 **	−0.845 **	0.914 **
	Sig. (two-tailed)	0.000	0.000	0.000
	N	47	47	47

** The correlation was significant at the 0.01 level (two-tailed). CEESUMTOT: sum of CEECCA scores for the entire sample. BOSTON_SUM: sum of Boston test subtest scores for the entire sample. NANDACDSUM: sum of the DCs of the NANDA-I label present in the sample. NOCINDSUM: sum of the NOC outcome indicator scores for the entire sample.

**Table 9 ijerph-20-03935-t009:** Cohen’s *κ* correlation coefficients between nurses. Areas in the CEECCA.

Areas in the CEECCA		Functionality Concordance (%)	Dysfunctionality Concordance (%)	Total Concordance (%)	Cohen’s *κ* Value	Sig. *p*
Nurse (a)	Nurse (b)					
Verbal expression: Conversational speech.		66.0	29.8	95.8	0.90 ***	<0.001
Verbal expression: Descriptive speech.		40.4	57.4	97.8	0.96 ***	<0.001
Verbal expression: Naming objects verbally.		55.3	40.4	95.7	0.91 ***	<0.001
Verbal expression: Naming actions verbally.		57.4	31.9	89.3	0.77 **	<0.001
Written expression: Writing name and surname(s).		48.9	42.6	91.5	0.83 ***	<0.001
Written expression: Naming objects in writing.		36.2	57.4	93.6	0.87 ***	<0.001
Written expression: Naming actions in writing.		25.5	66.0	91.5	0.80 **	<0.001
Expressing actions through pictograms.		83.0	8.5	91.5	0.62 **	<0.001
Expressing emotions through pictograms.		68.1	19.1	87.2	0.67 **	<0.001
Auditory comprehension of words.		83.0	17.0	100	1.00 ***	<0.001
Auditory comprehension of sentences.		42.6	29.8	72.4	0.44 *	0.003
Auditory comprehension of verbal commands.		44.7	29.8	74.5	0.48 *	0.001
Reading comprehension of words.		55.3	36.2	91.5	0.82 ***	<0.001
Reading comprehension of sentences.		36.2	55.3	91.5	0.82 ***	<0.001

Level of agreement according to Landis and Koch [39]: <0.20 slight; 0.21–0.40 fair; 0.41–0.60 moderate *; 0.61–0.80 substantial **; 0.81–1.00 almost perfect ***.

**Table 10 ijerph-20-03935-t010:** Intra-nurse Cohen’s *κ* correlation coefficients. Nurse (a) at baseline versus nurse (a) at one month.

Areas in the CEECCA		Functionality Concordance (%)	Dysfunctionality Concordance (%)	Total Concordance (%)	Cohen’s *κ* Value	Sig. *p*
Nurse (a) at Baseline	Nurse (a) at One Month					
Verbal expression: Conversational speech.		63.6	24.2	87.8	0.71 **	<0.001
Verbal expression: Descriptive speech.		39.4	48.5	87.9	0.76 **	<0.001
Verbal expression: Naming objects verbally.		48.5	36.9	85.4	0.69 **	<0.001
Verbal expression: Naming actions verbally.		60.6	33.3	93.9	0.87 ***	<0.001
Written expression: Writing name and surname(s).		57.6	33.3	90.9	0.81 ***	<0.001
Written expression: Naming objects in writing.		42.4	48.5	90.9	0.82 ***	<0.001
Written expression: Naming actions in writing.		30.3	63.6	93.9	0.86 ***	<0.001
Expressing actions through pictograms.		87.9	3.0	90.9	0.35	0.038
Expressing emotions through pictograms.		69.7	6.1	75.8	0.18	0.287
Auditory comprehension of words.		84.8	9.1	93.9	0.72 **	<0.001
Auditory comprehension of sentences.		42.4	24.2	66.6	0.31	0.073
Auditory comprehension of verbal commands.		48.5	33.3	81.8	0.63 **	<0.001
Reading comprehension of words.		63.6	27.3	90.9	0.79 **	<0.001
Reading comprehension of sentences.		42.4	39.4	81.8	0.64 **	<0.001

Level of agreement according to Landis and Koch [39]: <0.20 slight; 0.21–0.40 fair; 0.41–0.60 moderate; 0.61–0.80 substantial **; 0.81–1.00 almost perfect ***.

**Table 11 ijerph-20-03935-t011:** Intra-nurse Cohen’s *κ* correlation coefficients. Nurse (b) at baseline versus nurse (b) at one month.

Areas in the CEECCA		Functionality Concordance (%)	Dysfunctionality Concordance (%)	Total Concordance (%)	Cohen’s *κ* Value	Sig. *p*
Nurse (b) at Baseline	Nurse (b) at One Month					
Verbal expression: Conversational speech.		64.3	35.7	100	1.00 ***	<0.001
Verbal expression: Descriptive speech.		42.9	50.0	92.9	0.86 ***	0.001
Verbal expression: Naming objects verbally.		57.1	42.9	100	1.00 ***	<0.001
Verbal expression: Naming actions verbally.		57.1	35.7	92.8	0.85 ***	0.001
Written expression: Writing name and surname(s).		35.7	57.1	92.8	0.85 ***	0.001
Written expression: Naming objects in writing.		28.6	64.3	92.9	0.84 ***	0.001
Written expression: Naming actions in writing.		14.3	71.4	85.7	0.58 *	0.031
Expressing actions through pictograms.		78.6	21.4	100	1.00 ***	<0.001
Expressing emotions through pictograms.		50.0	28.6	78.6	0.55 *	0.036
Auditory comprehension of words.		71.4	28.6	100	1.00 ***	<0.001
Auditory comprehension of sentences.		42.9	42.9	85.8	0.71 **	0.008
Auditory comprehension of verbal commands.		42.9	28.6	71.5	0.43 *	0.094
Reading comprehension of words.		42.9	50.0	92.9	0.86 ***	0.001
Reading comprehension of sentences.		14.3	78.6	92.9	0.76 **	0.003

Level of agreement according to Landis & Koch [39]: <0.20 slight; 0.21–0.40 fair; 0.41–0.60 moderate *; 0.61–0.80 substantial **; 0.81–1.00 almost perfect ***.

## Data Availability

The data presented in this study are available upon request from the corresponding author. The data are not publicly available due to privacy/ethical restrictions.

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
