# Peer review of "Psychometric Testing of the CEECCA Questionnaire to Assess Ability to Communicate among Individuals with Aphasia"

_ijerph, 2023, doi:10.3390/ijerph20053935_

Round 1

Reviewer 1 Report

Article too long contains 40 pages, the content is not ordered.

The authors present extensive results that could be published in several shorter articles

The article needs  major revision.  In the introduction section, the abbreviation NANDA-I, NOC SNLs, should be explained

Section 2 Methodology should consider presenting the test method according to the PRISMA guidelines,

E.g. study design. The presentation of the methodology in the form of tables is definitely not clear in the message.

There is no information on the statistical analysis performed

There are no criteria for inclusion of the research group according to clinical parameters: age, sex, disease entities

Same with the exclusion criteria

In the results section, tables with statistical significance analysis should be presented

Tables 5-9 can be replaced with a verbal description,

Tables 10-12, 15-16 no information on the statistical tests used no information on the level of statistical significance and power of the statistical test used.

Tables 17-18 are not clear in the message

In the discussion section, Tables 19 and 20 distort the structure of the scientific article.

In the article standard, it is not allowed to present discussions in the form of a table, tables should be presented in the results section

Add limitations section 

Conclusions are too general.

Reviewer 2 Report

This is a study based on the necessity for adequate assessment of the communicative abilities of aphasia patients. The aim of this manuscript is to present a new psychometric testing of the CEECCA questionnaire, that intends  to assess the ability to communicate among individuals with aphasia. The CEECCA is intended to assess the communicative abilities among patients with aphasia based on areas of interest for care.

Although it is nicely presented, I am not at all convinced about the scientific soundness and the overall merit of such a study. Apart from that, your sample is very restricted and the time period that was under investigation is limited. This weakens even more the validity of your study, even if we suppose that these are preliminary results.

Reviewer 3 Report

The article submitted for review discusses research area that is very difficult and raises a lot
of questions and doubts. One of them refers to the correctness and credibility of answers provided by people with aphasia.

The authors might want to consider revising the following:

1.    It might be judicious to abandon the research hypothesis.

2.    It might be wise to simplify the purpose of the study by excluding the sample description.

3.    It might be worth specifying how the cognitive level of the research participants was assessed.

4.    The information should be provided whether the study has received approval of the relevant bioethics committee or whether it was conducted in accordance with the Declaration of Helsinki.

5.    It might be advisable to remove the tables from the Discussion section.

6.    It might be judicious to refrain from referencing certain works, especially those that were published in the 1990s and at the beginning of the 21st century.

Round 2

Reviewer 1 Report

The authors made more improvements in the Results, Discussions and Limitations sections

They still have not specified the methodology - the selection of the study group raises doubts as to the reliability of the study

The articles presented by the authors as validation -  they are published in local journals, not in IF journals, which confirms the lower credibility of the results obtained

I do not agree with the authors that the PRISMA scheme applies only to review articles.

Author Response

The authors thank you again for your contributions and we believe that they improve the quality of the manuscript.

Responses

The authors made more improvements in the Results, Discussions and Limitations sections

They still have not specified the methodology - the selection of the study group raises doubts as to the reliability of the study

The methodology is specified, the protocols used are cited, each of the phases of the process are described in the data collection and statistical analysis sections. You can find similar methodology in:

Park, D.-I. (2021). Development and Validation of a Knowledge, Attitudes and Practices Questionnaire on COVID-19 (KAP COVID-19). International Journal of Environmental Research and Public Health, 18(14), 7493. https://doi.org/10.3390/ijerph18147493

Some clarification has been added to the inclusion criteria.

The articles presented by the authors as validation -  they are published in local journals, not in IF journals, which confirms the lower credibility of the results obtained

The authors understand their suggestion, but believe that the reliability of the results of the studies presented should not only be judged based on the journals in which they have been published, but on their appropriate methodology. We insist on the limited publication of similar studies designed with samples composed of people with aphasia.

The systematic review used to justify the sample size is published in the Journal of Neurology.

El Hachioui, H., Visch-Brink, E. G., de Lau, L. M. L., van de Sandt-Koenderman, M. W. M. E., Nouwens, F., Koudstaal, P. J., & Dippel, D. W. J. (2017). Screening tests for aphasia in patients with stroke: A systematic review. Journal of Neurology, 264(2), 211-220. https://doi.org/10.1007/s00415-016-8170-8

I do not agree with the authors that the PRISMA scheme applies only to review articles.

The authors understand the relevant utility of the PRISMA scheme to assess the methodology used in the studies and, therefore, to assess the reliability of the results. However, the PRISMA 2020 statement has been designed primarily for systematic reviews of studies evaluating the effects of health interventions, regardless of the design of the included studies (Page et al., 2021). The authors understand that the checklist items are applicable to other reviews of scientific literature and could have been used in the design phase and in the discussion phase of this manuscript. The PRISMA scheme flowchart could also have been used to support the search strategies that were published in the article dedicated to the design and content validation of the CEECCA questionnaire (Martín-Dorta et al., 2021). In addition, the authors of this article understand the recommendation of the authors of the PRISMA strategy that reviewers and editors of journals support and publicize it (Page et al., 2021). We believe that this tool significantly contributes to the quality of systematic reviews, as its authors point out; however, this is not a frequently used tool in studies of design and validation of health questionnaires.

Martín-Dorta, W.-J., Brito-Brito, P.-R., & García-Hernández, A.-M. (2021). Development and Content Validation of the CEECCA Questionnaire to Assess Ability to Communicate among Individuals with Aphasia Based on the NANDA-I and NOC. Healthcare (Basel, Switzerland), 9(11), 1459. https://doi.org/10.3390/healthcare9111459

Page, M. J., McKenzie, J. E., Bossuyt, P. M., Boutron, I., Hoffmann, T. C., Mulrow, C. D., Shamseer, L., Tetzlaff, J. M., & Moher, D. (2021). Updating guidance for reporting systematic reviews: Development of the PRISMA 2020 statement. Journal of Clinical Epidemiology, 134, 103-112. https://doi.org/10.1016/j.jclinepi.2021.02.003

Page, M. J., McKenzie, J. E., Bossuyt, P. M., Boutron, I., Hoffmann, T. C., Mulrow, C. D., Shamseer, L., Tetzlaff, J. M., Akl, E. A., Brennan, S. E., Chou, R., Glanville, J., Grimshaw, J. M., Hróbjartsson, A., Lalu, M. M., Li, T., Loder, E. W., Mayo-Wilson, E., McDonald, S., … Moher, D. (2021). The PRISMA 2020 statement: An updated guideline for reporting systematic reviews. PLOS Medicine, 18(3), e1003583. https://doi.org/10.1371/journal.pmed.1003583

Kind regards,

The authors

Reviewer 2 Report

I have carefully studied your reply to my comments. Iam not convinced that this study fulfills the criteria for publication to this Journal. 

Author Response

Responses

I have carefully studied your reply to my comments. Iam not convinced that this study fulfills the criteria for publication to this Journal. 

The authors appreciate your contribution, however, a more specific analysis would be necessary in order to make improvements to the manuscript.

We have made some clarifications in the writing.

Regarding the English translation of the manuscript, it has been translated by a professional translator specialising in scientific papers and by a native English medical translator and editor. Please find the language editing certificate attached.

Kind regards,

The authors